# ZerOmics: Toward General Models for Single-Cell Analysis with Instruction Tuning

## Abstract

A variety of analysis tasks in single-cell (SC) multi-omics are crucial for precision medicine and clinical research. To address these tasks, existing methods are typically pre-trained on large-scale datasets to obtain general representations, followed by fine-tuning on specific tasks and labeled datasets. However, their task-specific heads often lack generalizability, significantly limiting performance in zero-shot scenarios. Inspired by the success of large language models (LLMs), we propose ZerOmics, the first zero-shot method that guides LLMs to perform various SC tasks without relying on specific downstream data. To enable LLMs to establish a correct and comprehensive understanding of SC data, ZerOmics employs a dual-alignment strategy. Specifically, ZerOmics aligns SC expression data with the well-organized gene corpus, thereby generating robust SC embeddings. These embeddings are then incorporated into instructions designed for various SC analysis tasks to tune the LLM, achieving alignment between SC data and the LLM. Extensive experiments across various sequencing technologies and tissues demonstrate that ZerOmics provides a comprehensive and general solution for SC analysis, achieving performance comparable to or even surpassing the state-of-the-art (SOTA) supervised and fine-tuned methods.

## 1 Introduction

Large language models (LLMs) recently have emergent abilities in understanding and reasoning, demonstrating the potential across a variety of applications. Increasing research shows that LLMs achieve expert-level performance in addressing problems from the natural sciences. For instance, LLMs have been successfully applied to drug molecule design (Li et al., 2024; Liu et al., 2024), protein structure prediction (Madani et al., 2023; Jin et al., 2024; Xiao et al., 2024), and reasoning about physical formulas (Ding et al., 2023). However, in the biomedical field, current LLMs are often confined to protein sequence analysis, neglecting the rapidly expanding single-cell (SC) multi-omics data.

As an emerging technology, SC multi-omics data provides valuable opportunities for comprehensive analysis of biological heterogeneity at multiple levels, including transcriptomics and epigenomics, within individual cells (Valous et al., 2024). A classic SC dataset is organized as a matrix $X \in \mathbb{R}^{N \times G}$, where $X_{i,j}$ represents the expression read counts of the $j$-th gene in the $i$-th cell, and $N$ and $G$ denote the number of cells and genes, respectively. Machine learning models trained on $X$ can accurately identify diseased cells (Sh et al., 2022), annotate cell types (Yang et al., 2022; Cui et al., 2024), and infer cell pathways (Subramanian et al., 2005; Fan et al., 2024), presenting unprecedented opportunities for advancements in clinical research and targeted therapy development (Aevermann et al., 2018).

SC data from different sources show differences caused by non-biological factors, including experimental conditions and instrument errors, leading to domain shifts (Zhao et al., 2020). To achieve domain adaptation in SC analysis process, models based on the "pre-training & fine-tuning" paradigm in Figure 1(a) have been widely adopted. Techniques like scBERT (Yang et al., 2022), Geneformer (Theodoris et al., 2023), and scGPT (Cui et al., 2024) are inspired by natural language processing workflows, treating the gene expression matrix $X$ as a "term frequency" matrix, regarding each cell as a sentence, and each gene as a word. These methods aim to pre-train robust models using auxiliary tasks on abundant unlabeled SC data. However, the general cell embeddings generated

Figure 1: Comparing (a) pre-training and fine-tuning paradigm with (b) ZerOmics paradigm.

by these methods are typically task-agnostic, necessitating fine-tuning on task-specific datasets with high-quality labels for optimal performance. When the fine-tuning dataset is limited in size, the capacity of these methods to tailor embeddings to particular task requirements is significantly diminished. Furthermore, the effectiveness of these models is undermined by inadequately designed task-specific heads, which fail to fully exploit the potential of the general embeddings, thereby compromising the overall efficacy of the model.

To reduce the heavy reliance of the existing methods on downstream task-specific heads and labeled datasets, we propose a novel framework, ZerOmics, as shown in Figure 1(b). Inspired by the breakthrough of LLMs in zero-shot scenario (Wei et al., 2022), ZerOmics unifies various SC tasks with text-based question answering and directly solves them with the help of LLM's excellent reasoning ability. Specifically, it employs a dual-alignment strategy: semantic alignment between SC expression and gene corpus, and between SC embeddings and the LLM. First, the SC expression $X$ is integrated with text embeddings extracted from gene text summaries, resulting in robust SC embeddings after large-scale pre-training. These embeddings are then incorporated into instructions designed for various SC tasks to tune the LLM, aligning with the LLM semantic space. In this way, after instruction tuning the LLM on multiple tasks, ZerOmics can successfully handle unseen datasets and tasks without any additional training. In brief, our contributions are summarized as:

- We propose ZerOmics, a novel framework that departs from the traditional pre-training and fine-tuning paradigm, establishing the first general model based on LLMs for SC multi-omics analysis.

- We introduce an innovative dual-alignment strategy that aligns SC gene expression data with a structured gene corpus, and SC embeddings with the LLM, enabling LLMs to establish a comprehensive interpretation of SC data.

- Extensive experiments across various sequencing technologies and tissues validate that ZerOmics achieves performance comparable to even exceeding that of state-of-the-art (SOTA) supervised and fine-tuned methods.

## 2 RELATED WORK

### 2.1 MULTI-MODAL INSTRUCTION TUNING FOR LLMS

In recent studies, tuning LLMs with multi-modal instructions has gathered great attention as an efficient strategy for enabling LLMs to comprehend information across diverse modalities. TEA-GLM (Wang et al., 2024) and GraphGPT (Tang et al., 2024) leverage the graph instruction paradigm to align graph representations with the LLM token embeddings, achieving zero-shot graph learning and guiding LLMs to comprehend graphs' inherent structural information. mPLUG-Owl2 (Ye et al., 2024) designs a modality-adaptive module to project both textual and visual information into a shared semantic space, achieving cross-modality interaction and preserving modality-specific features simultaneously. AnyRef (He et al., 2024) extracts features from images, bounding boxes, and audio followed by mapping them to the LLM token space, enabling flexible referring beyond single textual descriptions. Despite the variety of data types in biomedical scenarios (e.g., SC tabular matrix, spatial transcriptomics, spatial gene expression images), none of the studies explored how to

integrate biological information and natural language text into a unified space from the perspective of multimodal instruction tuning, so that LLM can directly answer various SC analysis tasks.

## 2.2 FOUNDATION MODELS FOR SINGLE-CELL ANALYSIS

LLMs have achieved remarkable breakthroughs across various domains, exemplified by models such as GPT-4 (Achiam et al., 2023). Inspired by them, many transformer-based foundation models such as scBERT (Yang et al., 2022), Geneformer (Theodoris et al., 2023), scGPT (Cui et al., 2024), and scFoundation (Hao et al., 2024) conceptualize SC gene expression as sentences within the language model, merely mimicking LLMs' training strategy to build non-language models for SC data. They pre-train the robust SC embeddings on extensive unlabeled single-cell RNA sequencing (scRNA-seq) datasets, followed by supervised fine-tuning across various downstream tasks.

However, gene expression data are often affected by the non-biological factors and the information they present is not as stable as text (Du et al., 2019). Therefore, pioneers in SC analysis, such as BioTranslator (Xu et al., 2023) and LangCell (Zhao et al., 2024), try to incorporate text data to make the extracted embeddings more general and robust. Both of them integrates the textual descriptions with biological expression data through pre-trained language models. In particular, LangCell constructs a cell-text dataset utilized to facilitate model pre-training via four tasks grounded in masking and contrastive learning strategies on cell-cell, cell-text pairs, demonstrating initial "representation" abilities in zero-shot and few-shot scenarios. Another attempt is Cell2Sentence (Levine et al., 2024), which converts each cell into a sentence of gene names ranked by descending expression abundance, and then directly uses the language model for representation. This approach retains only a minimal level of expression data, resulting in insufficient recognition of cell-specific information.

Therefore, how to effectively integrate expression data that presents cell-specific information and relatively more stable and consistent text data is still a problem that needs to be explored. In addition, although the embeddings extracted by many foundation models are robust in the zero shot setting, they still require task-specific heads to handle downsteam tasks.

## 3 METHODOLOGY

In this section, we introduce the novel framework ZerOmics, designed for zero-shot learning for various SC data analysis tasks. As shown in Figure 2, the ZerOmics comprises two principal training stages: (a) pre-training the SC Model to align SC expression profiles with the gene corpus, and (b) multi-modal and multi-task instruction tuning to align SC embeddings with the LLM semantic space. (c) After tuning, ZerOmics achieves zero-shot analysis in various SC downstream tasks without fine-tuning on labeled SC datasets.

### 3.1 SINGLE-CELL MODEL PRE-TRAINING

Considering the significant non-biological domain shifts often present in single-cell multi-omics data, and the limitations of SC expression matrices in accurately depicting cellular characteristics, ZerOmics incorporates a multi-modal mask learning paradigm for model pre-training. Our method synchronizes SC expression profiles with gene functions encapsulated in natural language, yielding semantically enriched, robust, and more distinctly characterizable single-cell embeddings, thus enhancing their utility in downstream analytical tasks.

**Single-cell expression embedding.** Single cell expression data often exhibit significant variation, including long-tail effects and domain shifts (Perez et al., 2022). To enhance the robustness of expression embeddings against domain shifts and improve computational efficiency (Yang et al., 2022), the elements in the expression matrix are binned into $\tilde{\boldsymbol{X}} \in \mathcal{B} = \{0, 1, 2, \cdots, N_{bins}\}^{N \times G}$ based on expression levels for tokenization, where $N_{bins}$ denotes the number of bins (see Appendix A.3 for details). Additionally, from a biological perspective, this operation emphasizes cell-specific information. Genes that are highly expressed in most cells, such as housekeeping genes, may exhibit lower expression levels in this context. In contrast, genes that are lowly expressed but crucial for identifying cell states, such as transcription factors, may exhibit higher expression levels (Theodoris et al., 2023). To improve model generalization, the random masking (RM) and random substitution (RS) strategy are employed (Kenton & Toutanova, 2019). Due to the sparsity of the expression

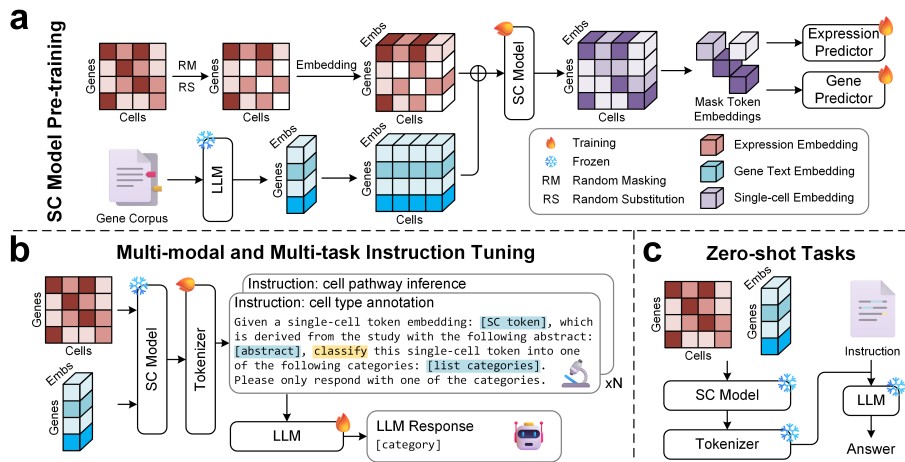

Figure 2: The overall framework of ZerOmics with single-cell data instruction tuning paradigm.

matrix, the effectiveness of RM is ensured by sampling the masking matrix $M \in \{0,1\}^{N \times G}$ as

$$M_{i,j} = \begin{cases} 1, & \text{if } X_{i,j} = 0, \\ m \sim \text{B}(1, 1 - p_m), & \text{otherwise,} \end{cases} \tag{1}$$

where $0 < p_m < 1$ denotes the mask proportion. The RS strategy replaces a certain proportion $0 < p_s < 1$ of binned values with other values in $\mathcal{B}$. After RM and RS, the matrix $\tilde{X}$ can be transformed into tokens $\tilde{X}^*$, then mapped into a learnable embedding space as

$$f_E(\tilde{X}^*) = Z_E \in \mathbb{R}^{N \times G \times d}, \tag{2}$$

where $d$ represents the dimension of the embeddings and $f_E$ is the learnable mapping.

**Gene text embedding.** Gene corpus can complement expression data and provide valuable opportunities to directly reveal cellular characteristics. Given a known gene, the corresponding item in the corpus summarizes the associated diseases, biological processes, and other genes in natural language (see Appendix A.2 for details). To encode the textual contents, a special item token is added at the end of each summary item, followed by being mapped into item-level embeddings using an autoregressive text model $f_T$, such as LLaMA (Touvron et al., 2023). Formulaically, the textual contents $C \in \mathbb{R}^{G \times L}$, where $L$ denotes the maximum length of text tokens, are encoded as

$$f_T(C) = Z_T \in \mathbb{R}^{G \times d}. \tag{3}$$

**Multi-modal mask learning for semantic alignment.** Cell expression and gene text embeddings reveal distinct levels of biomedical information within SC data. Using multi-modal learning to align their semantic spaces inspires the model to extract more comprehensive representations. For computational efficiency, ZerOmics utilizes the broadcasting to directly add $Z_T$ to $Z_E$, which is then encoded to the contextual SC embeddings that contain the gene functional semantics, as follows:

$$f_{SC}(Z_E \oplus Z_T) = Z_{SC} \in \mathbb{R}^{N \times G \times d}, \tag{4}$$

where $\oplus$ denotes the addition via the broadcasting and $f_{SC}$ represents the encoder-only SC Model. Effective generation of mask token embeddings should ensure they exhibit significant gene expression and textual information, enabling a single-layer network to accurately predict their original expression bins and corresponding gene summary items. Consequently, these mask token embeddings are transformed into probability distributions $p_E \in [0,1]^{N_{bins}}$, and $p_T \in [0,1]^G$ for gene expression bins and gene summary items using two independent predictors. Given the ground truth labels $y_E$ and $y_T$ for the bin and gene items, the loss function in mask learning is defined as

$$\mathcal{L}_M = \frac{1}{2} \left[ \text{CE}(p_E, y_E) + \text{CE}(p_T, y_T) \right], \tag{5}$$

where CE denotes the cross-entropy loss. Finally, via freezing the pre-trained $f_T$ and optimizing $\mathcal{L}_M$ using gradient descent, the optimal $f_E$ and $f_{SC}$ can be obtained.

### 3.2 MULTI-MODAL MULTI-TASK INSTRUCTION TUNING WITH SINGLE-CELL DATA

We leverage the instruction tuning paradigm (Wei et al., 2022) to enhance the adaptability of LLMs in capturing biomedical information from SC data. Firstly, SC embeddings are generated from a pre-trained SC Model, followed by being mapped into the LLM semantic space using a tokenizer and incorporated into various task-related instructions. By tuning the LLM with multi-task instructions to produce responses that increasingly resemble real sentences, the SC embeddings gradually align with the LLM, achieving superior performance across various downstream tasks.

**Single-cell token embeddings for LLM.** Considering that SC embeddings and LLM token embeddings involve different semantic information, a linear tokenizer $f_{token}$ is employed to transform the SC embeddings to the fixed length of SC token embeddings for LLM individually as

$$f_{token}(\boldsymbol{z}_i) = \boldsymbol{h}_i \in \mathbb{R}^{K \times d}, \tag{6}$$

where $\boldsymbol{z}_i \in \boldsymbol{Z}_{SC}$ denotes the extracted embedding of a cell by expression and gene text information, and $K$ is the length of tokens.

**Instruction design.** The instructions for various tasks in ZerOmics are uniformly constructed with three components: (1) single-cell information about the cell to be analyzed, (2) dataset information, and (3) task description.

First, the previously mentioned SC token embedding $\boldsymbol{h}_i$ (also [SC token] in instructions) contains comprehensive single-cell expression and gene function information, effectively representing the cell to be analyzed and serving as the first component.

To prevent the LLM from drawing one-sided conclusions by solely focusing on the current SC token, the data source information is incorporated for each cell. Providing additional context through dataset information enables the LLM to understand important factors, such as experimental conditions, technical platforms, and species—related to cell expression, significantly enhancing its ability to identify single-cell patterns across datasets. For simplicity, the abstracts of the papers that produced these datasets ([abstract] in instructions) serve as the second component, as they offer detailed research objectives, experimental settings, and other relevant information in natural language, effectively conveying dataset information (Tang et al., 2024).

Finally, the instruction tuning tasks encompass three representative and important SC analysis tasks: cell type annotation, rare cell identification, and tumor cell discovery (see Appendix B.1). To enhance the accuracy of the model response, the task description component contains both an imperative or question sentence matching the task and a set of answer candidates for the LLM. For example, for cell type annotation, the task description is structured as follows: `classify this single-cell token into one of the following categories: [list categories]. Please only respond with one of the categories.` Here, `[list categories]` is answer candidates related to the datasets. This design effectively guides LLMs to use the provided data to infer answers rather than memorizing them based on the dataset (Wang et al., 2024). Additionally, using candidate answers encourages the model to compare and contrast different options, strengthening its decision-making process and reducing the likelihood of generating incoherent responses (Kim et al., 2024).

We design three different instruction templates for each task to guide the LLM to truly understand the various tasks rather than memorize the instructions (Wei et al., 2022). Instruction examples can be found in Appendix E.

**Tuning strategy.** We collect single-cell multi-omics datasets containing 91.5M cells in total and a gene corpus containing 43.8K genes for instruction tuning, where the ground truth labels of three tasks are artificially constructed (see Appendix A.3 for details). Given a single epoch of tuning, each training sample is randomly assigned with one task and one corresponding instruction template. ZerOmics performs supervised tuning for the LLM based on the LoRA (Hu et al., 2022) in a random batch manner. To alleviate the catastrophic forgetting in the multi-task conditions (Wang et al., 2023), the LLM is tuned with the mixture of universal and task-specific LoRAs as Figure 3. In the forward process, the token embeddings from the pre-trained LLM and all LoRAs are summed to produce the output. Then, the cross-entropy loss between the probability distribution of the

output over the vocabulary and the ground truth one-hot label is computed for backpropagation (BP). However, only the universal LoRA and the LoRA for the current task are updated via gradient descent. This strategy aims to train a universal LoRA for domain transfer from general text to biomedical information, alongside a series of LoRAs specialized in SC analysis tasks. Moreover, the integration of LoRAs has demonstrated its superiority in zero-shot scenarios (Zhengmao et al., 2023).

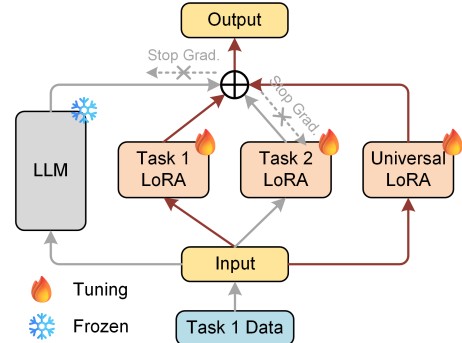

Figure 3: Tuning the LLM with the mixture of universal and task-specific LoRAs. For $n$ instruction tasks, there are $n$ task-specific LoRAs. Here, we take $n = 2$ as an example.

## 4  EXPERIMENTAL RESULTS

In this section, we extensively evaluate ZerOmics compared to other supervised, fine-tuned methods. We aim to investigate the following research questions: **RQ1:** How does ZerOmics perform in cross-dataset zero-shot learning scenarios? **RQ2:** How does ZerOmics perform in cross-task zero-shot learning scenarios? **RQ3:** How do the key components of ZerOmics influence the performance? **RQ4:** How do the parameters of LLM in ZerOmics affect the performance?

### 4.1  EXPERIMENTAL SETTINGS

We collect single-cell multi-omics datasets containing 91.5M cells from different species, tissues and diseases. To assess the performance in zero-shot setting, SC datasets are categorized into: (1) large-scale datasets for ZerOmics pre-training and instruction tuning (2) training and fine-tuning datasets for baseline models (unavailable for ZerOmics in zero-shot setting, partially available for ZerOmics in few-shot settings) (3) 9 held-out evaluation datasets from various research and labs, used only for testing performance (see Appendix A for details). On these unseen held-out datasets, three SC tasks (cell type annotation, rare cell identification, and tumor cell discovery) are selected for evaluating the performance under cross-dataset zero-shot setting; a unseen task, cell pathway inference, are selected for evaluating the performance under cross-task zero-shot setting (see Appendix B for details). ZerOmics employs the pre-trained LLaMA2-13B as the default LLM. Detailed information about the benchmark methods and other implementation details can be found in Appendix C. All experiments are conducted on 8 NVIDIA A100 (80G) GPUs.

### 4.2  OVERALL PERFORMANCE COMPARISON (RQ1)

**Cell type annotation (CTA).**  We evaluate the CTA performance of ZerOmics across four datasets: 10x scRNA-seq human peripheral blood mononuclear cells (PBMC68K), Smart-Seq2 human pancreas (Pancreas), 10x scATAC-seq bone marrow mononuclear cells (BMMC), and MERFISH mouse primary motor cortex (MOP). Eight competing methods are selected for comparison with ZerOmics. Among them, LangCell is the only baseline method with zero-shot capability, and it also provides a fine-tuned version named LangCell-CE. The overall performance is summarized in Table 1. Due to the meticulously annotated labels, supervised methods such as Seurat demonstrate excellent performance across most datasets. Nevertheless, four fine-tuning methods achieve performance comparable to or surpassing Seurat only on PBMC68K and Pancreas, while significant disparities are observed in the other two datasets. This variation is attributed to the absence of 10x scATAC-seq or MERFISH sequencing data in their pre-training datasets. Additionally, the limited generalization ability of the pre-trained models impedes their performance even after fine-tuning on new datasets. Among zero-shot methods, while LangCell performs comparably to the supervised method scJoint, it also shows a rapid performance decline on BMMC, similar to its fine-tuning version, LangCell-CE. In contrast, ZerOmics achieves the best or second-best performance across most datasets. Notably, even without leveraging the spatial location information in MOP, its performance is only slightly inferior to that of Seurat. In summary, the above results not only indicate that ZerOmics, as a zero-shot method, has achieved state-of-the-art (SOTA) in the CTA task, but also suggest that LLMs are capable of comprehending cell-type information in SC datasets.

Table 1: Results of cell type annotation. Acc and F1 denote the accuracy and macro F1-score (as %) respectively. The best results are marked as **bold**. "–" indicates the method can't handle the experiment, while red indicates the method isn't suitable for the experiment but is still used.

| Paradigms | Methods | PBMC68K | | Pancreas | | BMMC | | MOP | | Avg. | |
|---|---|---|---|---|---|---|---|---|---|---|---|
| | | Acc↑ | F1↑ | Acc↑ | F1↑ | Acc↑ | F1↑ | Acc↑ | F1↑ | Acc↑ | F1↑ |
| Supervised | scJoint | 61.52 | 56.10 | 69.75 | 57.58 | 63.31 | 50.52 | 50.92 | 41.04 | 61.37 | 51.31 |
| | Spatial-ID | – | – | – | – | – | – | 82.54 | 73.79 | 82.54 | 73.79 |
| | Seurat v5 | 74.91 | 73.55 | 79.76 | 70.49 | 78.31 | **80.52** | **85.80** | **82.28** | 79.70 | 76.71 |
| Fine-tuning | scBERT | 75.74 | 67.34 | 69.21 | 67.03 | 17.09 | 9.25 | – | – | 54.01 | 47.87 |
| | Geneformer | 83.94 | 74.05 | 65.71 | 62.36 | 20.21 | 11.38 | – | – | 56.62 | 49.36 |
| | LangCell-CE | 85.22 | **76.38** | 80.61 | 72.73 | 19.48 | 11.94 | – | – | 61.77 | 53.68 |
| | scGPT | 84.48 | 75.39 | 70.76 | 68.03 | 67.18 | 60.93 | – | – | 74.14 | 68.12 |
| Zero-shot | LangCell | 67.07 | 53.51 | 68.18 | 53.55 | 10.43 | 6.67 | – | – | 48.56 | 37.91 |
| | ZerOmics | **85.56** | 74.70 | **86.59** | **78.34** | **79.09** | 75.17 | 83.68 | 80.72 | **83.73** | **77.23** |

Table 2: Results of rare cell identification. F1 and $\kappa$ denote the F1-score and Cohen Kappa score (as %) respectively.

Table 3: Results of tumor cell discovery. Acc and F1 denote the accuracy and F1-score (as %) respectively.

| Methods | PBMC68K | | Airway | |
|---|---|---|---|---|
| | $\kappa$↑ | F1↑ | $\kappa$↑ | F1↑ |
| MARS | 45.33 | 50.65 | 63.20 | 65.12 |
| scVI | 46.36 | 53.13 | 64.98 | 67.29 |
| scBalance | 63.67 | 64.56 | 69.28 | **71.02** |
| ZerOmics | **65.46** | **67.52** | **69.61** | 70.97 |

| Methods | CTC | | LungCancer | |
|---|---|---|---|---|
| | Acc↑ | F1↑ | Acc↑ | F1↑ |
| CopyKAT | 57.06 | 58.69 | 80.16 | 50.96 |
| CaSee | 86.67 | 90.98 | 60.69 | 36.92 |
| ikarus | 89.35 | **92.11** | 91.45 | 74.85 |
| ZerOmics | **92.81** | 91.53 | **92.13** | **85.62** |

**Rare cell identification (RCI).** We assess the performance of ZerOmics in RCI on two datasets: PBMC68K and 10x scRNA-seq mouse airway epithelium (Airway). Unlike CTA, RCI requires the model's sensitivity to recognize rare cells in the class imbalance scenario. Since fine-tuning models do not support this task, we select three classic supervised methods for evaluation. The results are reported in Table 2. We observe that ZerOmics achieves the best or second-best performance across all metrics. Due to its vast model parameters and training data, ZerOmics exhibits relatively balanced performance across different datasets, avoiding the scenario observed in MARS and scVI where one dataset performs well while another performs poorly.

**Tumor cell discovery (TCD).** We examine the performance of ZerOmics in TCD, a task more closely related to clinical medical research. This experiment involves two datasets: 10x scRNA-seq human circulating tumor cells (CTC) and Lung Cancer (LungCancer). Unlike CTA, TCD requires the model to distinguish cancer cells based on features such as mutations and gene expression heterogeneity inherent in the data itself. Therefore, we select three methods specialized for TCD, which usually model with prior knowledge about tumor expression features (such as CopyKAT, CaSee), or carcinogenic gene markers (e.g., ikarus). The comparison results are presented in Table 3. Among all baseline methods, ikarus achieves a significant advantage, indicating that incorporating carcinogenic gene information can effectively guide the model to recognize cancer. Coincidentally, the gene corpus used to train ZerOmics also includes an amount of natural language text describing the relationships between genes and corresponding diseases. Through the dual-alignment strategy, the LLM can well understand and respond to this information, as evidenced by ZerOmics achieving the best performance in all experiments only except for the F1-score in CTC.

## 4.3 CROSS-TASK ZERO-SHOT PERFORMANCE (RQ2)

We explore whether ZerOmics, as an instruction-tuned LLM, has the emergent ability to tackle the unseen task. Specifically, we evaluate its performance on the cell pathway inference (CPI), in both zero- and few-shot settings, comparing with the fine-tuned models, geneformer and LangCell-CE. This experiment involves two datasets: 10x scRNA-seq human dilated and hypertrophic cardiomyopathy (HDHC), and liver tissue (Liver). The results are summarized in Table 4.

Table 4: Results of cell pathway inference. AUROC and AUPRC denote the Area Under the Receiver Operating Characteristic Curve and Area Under the Precision-Recall Curve (as %) respectively. Prefixes a- and f- denote metrics calculated by different strategies.

| Paradigms | Methods | HDHC | | | | Liver | | | |
|---|---|---|---|---|---|---|---|---|---|
| | | a-AUROC↑ | f-AUROC↑ | a-AUPRC↑ | f-AUPRC↑ | a-AUROC↑ | f-AUROC↑ | a-AUPRC↑ | f-AUPRC↑ |
| Fine-tuning | Geneformer | 82.80 | 86.27 | 23.25 | 27.56 | 89.15 | 90.47 | 29.79 | 35.28 |
| | LangCell-CE | 89.33 | 89.45 | **31.23** | 35.08 | 92.15 | 92.53 | 34.65 | **36.63** |
| Zero-shot | ZerOmics | 80.27 | 83.92 | 12.74 | 19.15 | 85.14 | 86.56 | 21.15 | 27.67 |
| One-shot | | 80.08 | 83.60 | 14.46 | 20.57 | 87.60 | 88.58 | 24.61 | 27.24 |
| Five-shot | | 83.64 | 87.47 | 20.43 | 26.13 | 91.48 | 91.75 | 32.86 | 36.12 |
| Ten-shot | | **91.28** | **92.61** | 30.89 | **35.83** | **92.36** | **92.81** | **34.99** | 36.07 |

Cell pathways typically refer to a series of biological processes occurring within a cell, guided by interactions among genes (Subramanian et al., 2005). Coincidentally, the gene corpus used to train ZerOmics includes natural language descriptions of the biological processes in which each gene may be involved. Therefore, we observe that even without prior tuning on CPI, ZerOmics demonstrates remarkable emergent abilities, with performance only slightly behind the fine-tuned Geneformer. Subsequently, in few-shot settings, ZerOmics is instruction-tuned using 1, 5, and 10 samples on CPI, respectively. In the one-shot setting, the model's performance does not improve significantly and even declines in certain cases, such as the a-AUROC on HDHC. This could be due to the model incorrectly generalizing supervisory information from a single sample to all. Notably, after five-shot learning, ZerOmics shows a significant improvement on the Liver dataset, surpassing Geneformer and even approaching the superior LangCell. After ten-shot learning, ZerOmics surpasses the current SOTA method, LangCell, across most metrics, despite it undergoing extensive fine-tuning. These findings suggest that benefiting from the generalization capabilities of LLMs, ZerOmics often can effectively transfer their prior knowledge to unseen datasets and tasks.

## 4.4 Module Ablation study (RQ3)

To address RQ3, we remove or replace the main components of ZerOmics to assess their effectiveness. Specifically, we analyze the impact of gene text summaries, the SC Model, and the mixture of tuning-specific LoRAs. Comparison results of original and variant models are presented in Table 5. Additionally, all variant models are trained with changed settings from scratch.

**Text Summary.** Our primary concern is whether the model's strong performance attributes to memorize text summaries rather than using them to represent biological information within cells. To investigate the role of text summaries, we replace them with non-textual gene embeddings extracted from well-pretrained models, including Gene2vec (Du et al., 2019), Geneformer (Theodoris et al., 2023), and GeneCompass (Yang et al., 2024). While the original model outperforms all its variants, utilizing Geneformer and GeneCompass to embed genes demonstrates relatively stronger performance. This indicates that text summaries are not the decisive factor behind superior performance. However, as our model employs an LLM as the task processor, representing genes as text features often yields better results than pre-trained gene embeddings based on expression data.

**SC Model.** We address two primary concerns regarding the SC Model: (1) Can it be replaced with a simpler model? (2) Given the pre-trained SC Model, whether the SC tokenizer is redundant during the instruction tuning stage? To explore these questions, we first test GenePT (Chen & Zou, 2024), which generates gene text embeddings using an LLM and directly represents each cell by weighting these embeddings with expression values. Next, we compare the original model with a variant without the SC tokenizer (w/o SCT). Both variants show performance degradation; however, the change in performance for GenePT on the TCD task is insignificant. This suggests that combining text summaries is insufficient to fully capture the specificity of individual cells, resulting in suboptimal performance on cell-level tasks such as CTA. But this simple combination may be completely sufficient for TCD tasks. Furthermore, our results indicate that SCT remains an indispensable component of ZerOmics, primarily serving to align SC embeddings derived from general pre-training with the semantic space of the LLM.

Table 5: Module ablation study on three single-cell analysis tasks.

| Gene features | Extractors | CTA-PBMC68K | | RCI-Airway | | TCD-CTC | |
|---|---|---|---|---|---|---|---|
| | | Acc↑ | F1↑ | $\kappa$↑ | F1↑ | Acc↑ | F1↑ |
| Gene2vec | SC Model | 65.85 | 59.03 | 60.33 | 63.98 | 82.01 | 80.30 |
| Geneformer | | 77.37 | 70.69 | 64.18 | 66.22 | 85.29 | 82.60 |
| GeneCompass | | 79.71 | 73.51 | 64.56 | 67.89 | 87.24 | 83.69 |
| Text Summary | GenePT | 66.67 | 62.11 | 61.26 | 63.87 | 86.85 | 82.24 |
| | w/o SCT | 60.43 | 55.10 | 58.47 | 62.70 | 84.74 | 81.88 |
| Text Summary | w/o uLoRA | 76.92 | 67.62 | 65.46 | 67.67 | 84.18 | 79.21 |
| | w/o tLoRA | 66.43 | 58.63 | 59.64 | 58.57 | 83.30 | 80.63 |
| Text Summary | SC Model | **85.56** | **74.70** | **69.61** | **70.97** | **92.81** | **91.53** |

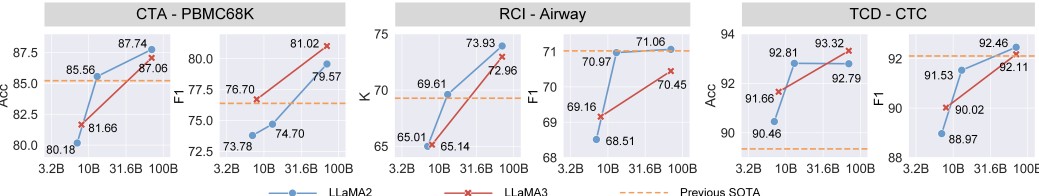

Figure 4: LLM scalability study on three single-cell analysis tasks.

**Mixture of LoRAs.** ZerOmics employs a mixture of LoRAs for tuning the LLM. To analyze their necessity, we construct two variants by removing the universal LoRA (uLoRA) and task-specific LoRAs (tLoRA), referred to as "w/o uLoRA" and "w/o tLoRA", respectively. Both variants exhibit significantly worse performance compared to the original model. Notably, in CTA and RCI tasks, the w/o uLoRA variant significantly outperforms w/o tLoRA, while the difference is less pronounced in the TCD task. We attribute this to the tendency of a single LoRA to prioritize optimization for simpler tasks like TCD, leading to the forgetting of information relevant to other tasks. Although task-specific LoRAs provide independent low-rank spaces, the resulting redundancy disrupts the model's overall performance.

## 4.5 LLM'S SCALABILITY STUDY (RQ4)

We further explore the scalability of the LLM within ZerOmics to address RQ4. Specifically, we investigate whether the parameter size and the pre-training quality of the LLM are capable of significantly changing the model performance. We assess instruction tuning performance across several models, including LLaMA2-7B, 13B, 70B, as well as LLaMA3-8B and 70B (Dubey et al., 2024), with the latter being pre-trained on larger, higher-quality datasets. Results are showcased in Figure 4. Concerning the parameter size, larger models consistently yield superior results, displaying an upward performance trend, except for the F1-score on Airway and Accuracy in the CTC task. Reducing the parameter size does not substantially diminish the performance compared to previous SOTA models. With regard to pre-training data, LLaMA3 does not consistently surpass LLaMA2. indicating that ZerOmics benefits more from larger models capable of capturing complex interactions between SC data and textual information rather than from merely expanding the LLM's knowledge base, which enables it to discern relationships within specific SC tasks effectively.

## 5 CONCLUSION

In this study, we introduce ZerOmics, a novel LLM-based framework designed for zero-shot single-cell multi-omics analysis. By simply inputting data and posing queries in natural language, ZerOmics intelligently addresses various biological tasks. We conduct comprehensive evaluations of its performance across diverse tissues, sequencing technologies, and species. The results confirm that ZerOmics achieves state-of-the-art performance, underscoring its potential to revolutionize single-cell multi-omics analysis.

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

# A DATASETS INFORMATION

## A.1 SINGLE-CELL DATASETS COLLECTION

We assemble single-cell multi-omics data from *Homo sapiens* and *Mus musculus*. We collect 1,465 relevant datasets primarily from the well-organized CELLxGENE database (Megill et al., 2021) at `https://cellxgene.cziscience.com/`, encompassing approximately 91.5 million cells and 900 cell types. These datasets are primarily categorized into single-cell transcriptomics and epigenomics, utilizing various sequencing technologies.

Single-cell transcriptomics datasets sequenced with different technologies are incorporated into the benchmark, including 10x scRNA-seq (Kolodziejczyk et al., 2015), Smart-seq (Goetz & Trimarchi, 2012), and Drop-seq (Macosko et al., 2015), etc. Additionally, spatial transcriptomics datasets at single (or sub-single) cellular resolution, such as Slide-seq (Rodriques et al., 2019), are also included. Single-cell epigenomics datasets are primarily sequenced with 10x scATAC-seq (Buenrostro et al., 2013) and snmC-seq (Luo et al., 2017). All the datasets are transformed into a high-dimensional matrix $\boldsymbol{X} \in \mathbb{R}^{N \times G}$, where $X_{i,j}$ denotes the expression read counts of the $j$-th gene in the $i$-th cell, and $N$ and $G$ denote the number of cells and genes, respectively. Most datasets include annotation label files for cell types. Datasets related to diseases include annotation label files for disease types, while those not associated with diseases are uniformly labeled as normal cells. Additionally, spatial transcriptomics datasets are organized into the same files, omitting their unique tissue domain type annotation and spatial coordinate annotation files.

## A.2 GENE CORPUS COLLECTION

We employ GeneCards (Rebhan et al., 1997) as gene corpus. GeneCards is an extensive textual database that offers a comprehensive view of the currently available genomic, proteomic, transcriptomic, genetic, and functional information on more than 350,000 known and predicted human genes, serving as an "encyclopedia" for biomedical research (Harel et al., 2009). The original data is available at `https://www.genecards.org/`. As examples, Table 6 lists some of the gene summary items from our gene corpus. Given a known gene, the corresponding item first summarizes its type and functional information, followed by its associated diseases, biological processes, and other genes, respectively. Additionally, to ensure that the gene names are meaningful and understandable, the HUGO Gene Nomenclature (Bruford et al., 2020) is uniformly used to provide the unique identifier gene symbols, which are usually abbreviations of gene functions. Genes not included in the HUGO Gene Nomenclature are often not studied in depth and are therefore discarded. Finally, we collect 43,850 gene summary items to constitute the gene corpus.

## A.3 DATA PREPROCESSING

ZerOmics preprocesses all the single-cell multi-omics datasets with a unified pipeline as follows:

**Gene list mapping.** After collecting the large-scale SC datasets and gene text corpus, we first transform their gene symbols to the HUGO Gene Nomenclature. Then we take the intersection of the genes in the gene text corpus and the SC datasets to form the one-to-one correspondence. Additionally, due to different sequencing protocols or different completeness, a single SC dataset often does not contain all the genes obtained by taking the intersection here. For each SC expression matrix, the expression values of these dropout genes are filled with zero. Thus, all the SC expression matrics are transformed to have the same column names (gene symbols).

**Quality control and normalization.** Low-quality cells, such as cells expressing few genes, are removed uniformly with Scanpy (Wolf et al., 2018). Here, we only keep the cells with over 200 genes expressed (i.e., the number of non-zero genes in expression vectors $> 200$) for subsequent training and analysis. To alleviate the differences in gene expression between different datasets due to sequencing depth, the total gene expression of each cell is normalized to 10,000 (i.e., library size). Considering the subsequent Binning process, these datasets are not transformed by log1p.

**Binning and tokenization.** To map continuous gene expression values to discrete tokens, non-zero gene expression values are divided into different bins according to their quantile values among

all non-zero values. Here, we set the number of bins as 10. So, for example, if a non-zero SC expression value is in the bottom 5% of all non-zero values, it's assigned to the 1st bin; if it is in the bottom 18%, it's assigned to the 2nd bin, and so on. ZerOmics does not focus on the specific SC expression value, but rather on its relative expression level. Therefore, in the pre-training stage of SC Model, all SC expression tokens include 12 types, namely 10 bin value tokens, zero value tokens, and the special mask token.

**Dataset splitting.** We categorize the SC datasets in our study based on their usages: (1) large-scale datasets used for Zeromics' pre-training and instruction tuning stage (2) training and fine-tuning datasets split from evaluation datasets used for baseline methods training and fine-tuning (unavailable for ZerOmics in zero-shot setting, partially available for ZerOmics in few-shot settings) (3) 9 held-out evaluation datasets, which are collected from diverse studies and labs only used for testing model performance. The detailed train-test split strategy between type 2 and type 3 datasets is similar to Langcell (Zhao et al., 2024), which can be also seen in each task description of Appendix B.1. Therefore, type 2 and type 3 datasets are split from the same evaluation datasets, sharing the same sampling conditions and sequencing processes, which approximately satisfy the same distribution assumption. In contrast, type 1 datasets originate from independent studies with different research objectives and sampling conditions, resulting in no correspondence with the held-out datasets. Consequently, type 3 dataset is designated as the unseen held-out datasets for evaluation.

# B    EVALUATION PROTOCOLS

## B.1    EVALUATION TASKS AND METRICS

**Cell type annotation (CTA).** CTA is the most classic multi-classification task in single-cell analysis. The conventional analysis process is to use the given SC dataset and cell type label file to train a classifier and use it to identify cell types in the same type of test set. For ZerOmics in the zero-shot setting, only the dataset needs to be provided, and it randomly assigns an instruction template to generate a text response for the cell type. Since supervised and fine-tuning methods need to be re-learned on downstream datasets, we further divide the evaluation datasets into training and test sets according to the common 2:1. We compare the classification results of all methods on the test set with the widely used metrics, accuracy and macro F1-score.

**Rare cell identification (RCI).** RCI is a special two-class classification task with imbalanced classes. The conventional analysis process is to use the given SC dataset and cell type label file to train a model. However, in the test set, the models compare the cells from the new dataset with the existing samples, and those samples that are difficult to be classified into existing types are regarded as rare cells. For ZerOmics in zero-shot setting, similar to the CTA process, the LLM directly generates binary classification text results. The evaluation datasets are split according to 2:1, and all methods generate results on the test set for comparison. F1-score and the Cohen Kappa score ($\kappa$) are employed for evaluation, where $\kappa$ is a metric that compares the prediction result with random guessing and is often used to detect imbalanced class samples.

**Tumor cell discovery (TCD).** TCD is also a two-class classification task. Unlike CTA and RCI, the conventional analysis process usually does not provide label files about tumor type of each cell, and the model needs to identify cancer cells in an unsupervised or weakly supervised manner. For ZerOmics in zero-shot setting, the LLM not only needs to respond to whether it is a tumor cell, but also needs to respond to what subtype of tumor it is. This strategy is mainly to enable ZerOmics to better understand and infer information related to cancer. The evaluation datasets are split according to 2:1, and all methods generate results on the test set for comparison. In addition, to ensure the consistency of the evaluation, the tumor subtypes predicted by ZerOmics are not considered, that is, the predicted results are uniformly treated as binary classification. Lastly, accuracy and F1-score are served as the evaluation metrics.

**Cell pathway inference (CPI).** In this study, CPI is a multi-category independent binary classification task. Specifically, we focus on the 41 hallmark pathways from the Broad Institute's Molecular Signatures Database (MSigDB) (Liberzon et al., 2011). For each cell, the model needs to determine

which one or several of these pathways it expresses. Since the ground truth of the pathway is often not directly provided, we use irGESA (Fan et al., 2024) to analyze the pathways that each cell may express in the evaluation datasets. Note that irGESA is not a predictive model, but a one-to-one matching model for the gene expression pattern of each cell and all known cell pathways. Geneformer and LangCell also need to use the generated labels for supervised fine-tuning, while ZerOmics in the zero-shot setting uses instructions to directly generate answers in a way similar to answering multiple-choice questions. The evaluation datasets are split according to 2:1, and all methods generate results on the test set for comparison. We employ the Area Under the Receiver Operating Characteristic Curve (AUROC) and Area Under the Precision-Recall Curve (AUPRC) for evaluation. Due to the complexity of the predicted labels, these two metrics are also calculated using two different strategies, namely average and flatten. Average (a-) metrics treat the 41 prediction results of each cell as independent samples for calculation, and then averages the AUROC or AUPRC results of each cell. While flatten (f-) metrics the 41 prediction results of all n cells as independent samples and calculates AUROC or AUPRC on these 41*n samples.

## B.2 EVALUATION DATASETS

We collect abundant benchmark datasets for evaluating the performance of ZerOmics across diverse tasks. The following introduction of each dataset summarizes the involved cell information and its application scenarios in this paper.

**PBMC68K.** The PBMC68K (Zheng et al., 2017) dataset is sourced from a healthy donor, consisting of the gene profiles of 68,450 peripheral blood mononuclear cells (PBMCs). The dataset encompasses eleven distinct cell types, including CD4+ T cells, CD8+ T cells, B cells, natural killer (NK) cells, CD14+ Monocytes, FCGR3A+ Monocytes, dendritic cells, memory cells, helper2 cells, and Megakaryocytes. Cells were processed on the 10x platform using the scRNA-seq technology. This dataset is utilized for the cell type annotation task in this paper. PBMC68K is used for CTA and RCI tasks in this study.

**Pancreas.** The Pancreas (Segerstolpe et al., 2016) dataset consists of 2,209 single cells compiled from human pancreatic islets, with samples collected from six healthy and four type 2 diabetes (T2D) donors. The dataset encompasses both endocrine and exocrine cells, a total of eight cell types: alpha, beta, gamma, delta, and epsilon endocrine cells, as well as acinar, ductal, and pancreatic stellate cells (PSCs). The cells were dissociated into single-cell suspensions and sorted using fluorescence-activated cell sorting (FACS), followed by RNA sequencing through the Smart-seq2 protocol. Pancreas is utilized for the CTA task in this study.

**BMMC.** The BMMC dataset referenced in (Granja et al., 2019) consists of 35,882 bone marrow mononuclear cells (BMMCs) collected from healthy donors. The dataset contains six cell types including progenitor cells, B cells, T cells, NK cells, monocytes, and dendritic cells. Cells were profiled on the 10x platform utilizing the single-cell assay for transposase-accessible chromatin using sequencing (scATAC-seq) technology. BMMC is utilized for the CTA task in this study.

**MOP.** The MOP (Zhang et al., 2023) dataset is a spatially resolved, molecularly defined cell atlas of an entire mouse brain. This dataset provides 338 major cell types over ten million cells across eleven major brain regions. The MOP dataset was collected by the Multiplexed Error-Robust Fluorescence In Situ Hybridization (MERFISH) technology, which is a spatial transcriptomics (ST) method that allows for gene expression profiling while preserving the spatial context of the cells within intact tissue sections. MOP is utilized for the CTA task in this paper.

**Airway.** The Airway (Montoro et al., 2018) dataset is profiled by scRNA-seq protocol and comprises 7,494 cells from mice. The dataset revealed seven cell types including basal cells, club cells, ciliated cells, tuft cells, neuroendocrine, goblet cells, and Foxi1+ pulmonary ionocyte cells. Airway is utilized for evaluating the RCI task in this study.

**CTC.** The CTC (Szczerba et al., 2019) dataset focuses on circulating tumor cells (CTCs) associated with white blood cells (WBCs), specifically neutrophils, in patients with breast cancer. This

dataset is profiled by Smart-seq2 and contains 357 cells. There is no annotation file about cell sub-types attached to the dataset, but it provides a binary annotation of whether it is a tumor or not. CTC is used for evaluating the TCD task.

**LungCancer.** The LungCancer (Qian et al., 2020) dataset consists of scRNA-seq profiles of 93,576 cells derived from patients with lung cancer. The dataset identified ten distinct cell types including tumors. LungCancer is used for evaluating the TCD task in this study.

**HDHC.** The HDHC (Chaffin et al., 2022) dataset consists of single-nucleus RNA sequencing (snRNA-seq) profiles of nearly 600,000 nuclei derived from the left ventricle samples of patients with hypertrophic cardiomyopathy and dilated cardiomyopathy (HDHC), and non-failing (NF) hearts. The dataset contains twenty-one distinct cell populations. HDHC is utilized for the CPI task in this study.

**Liver.** The Liver (MacParland et al., 2018) dataset is profiled by scRNA-seq technology, comprising 8,444 cells isolated from healthy human liver tissues obtained from five neurologically deceased donors. The dataset identified twenty cell types in total. Liver is used for the CPI task in this study.

## C    IMPLEMENTATION DETAILS

### C.1    ENVIRONMENTS

All experiments are conducted on eight NVIDIA Tesla A100 GPUs, each with 80 GB of memory. The various parameter versions of ZerOmics, along with a series of its variant models are trained using the PyTorch framework (Paszke et al., 2019), integrated with DeepSpeed (Rasley et al., 2020) and FlashAttention v2 (Dao, 2024) for optimized memory and computational efficiency. Gradient checkpointing is employed by default, a widely adopted technique in the PEFT (Parameter-Efficient Fine-Tuning) codebase (Mangrulkar et al., 2022), to further reduce memory overhead during training. Please note that eight A100 GPUs are not strictly necessary; they are mainly used to accelerate the training process through parallelization.

### C.2    BENCHMARK METHODS

In this section, we offer a concise overview of each benchmark method utilized in this study.

**Spatial-ID.** Spatial-ID (Shen et al., 2022) is a supervised benchmark for the CTA task, which is tailored for ST data. It first employs transfer learning to train a deep neural network (DNN) pre-trained on the reference scRNA-seq data. In the inference stage, it leverages a variational graph autoencoder (VGAE) to contain spot embeddings, followed by feeding them into the DNN-based classifier to generate pseudo-labels for each spot. The spatial embeddings are then combined with gene expression profiles to refine cell type predictions via a self-supervised learning approach.

**scJoint.** scJoint (Lin et al., 2022) is designed for the CTA task for both scRNA-seq and scATAC-seq data. Initially, it leverages annotated scRNA-seq data to guide the training process, transferring labels to unlabeled scATAC-seq data via constructing a KNN graph based on cell-cell similarities between two omics. scJoint simultaneously trains on labeled and unlabeled data, enabling effective label transfer and integration across heterogeneous multi-omics datasets.

**Seurat.** Seurat v5 (Hao et al., 2023) serves as one of the baseline methods for the unsupervised CTA task. It first builds a K-nearest neighbor (KNN) graph based on cell-cell similarities, followed by community detection to annotate cells into subtypes via the Louvain algorithm.

**scBERT.** scBERT (Yang et al., 2022), as the pioneer of large-scale pre-trained models for SC data, utilizes the performer (Choromanski et al., 2020) architecture with 6M parameters and pre-trained on over 1M unlabeled, preprocessed scRNA-seq samples. At the supervised fine-tuning stage, the pre-trained encoder is tuned with labeled task-specific scRNA-seq data to adapt distinct downstream tasks. We employ scBERT as a baseline method for the CTA task in this study.

**Geneformer.**  Geneformer (Theodoris et al., 2023) is pre-trained on nearly 30M scRNA-seq data samples, and adopts a CNN-based feature generator to learn cell representations, followed by a transformer-based entropy model. The model employs a latent array to manage sequence length to solve the gene compression problem. Finally, Geneformer applies transfer learning across various biological tasks. Geneformer is included as a benchmark for the CTA and CPI tasks in this study.

**scGPT.**  scGPT (Cui et al., 2024) serves as a benchmark method for CTA and RCI tasks in this paper. The model is generatively pre-trained on over 33M scRNA-seq samples, followed by supervised fine-tuning for specific downstream tasks, including CTA and batch integration.

**LangCell.**  Langcell (Zhao et al., 2024) is a recent work that integrates textual information with gene expression profiles during the pre-training stage. It builds a cell-text dataset utilized for pre-training vis four tasks, including mask gene modeling, cell-cell contrastive learning, cell-text contrastive learning, and cell-text matching to recognize the intricate relationships between SC and text modalities. We employ it as a benchmark method for the CTA task.

**scVI.**  scVI (Lopez et al., 2018), tailored for scRNA-seq data analysis, implements a completely probabilistic framework based on a hierarchical Bayesian model. The gene expression profiles are firstly encoded into low-dimensional embeddings and then decoded for computing posterior estimates of the distributional parameters for each gene in each cell. scVI is utilized as a benchmark for the RCI task.

**MARS.**  MARS (Brbić et al., 2020) is a baseline method for the RCI task. MARS firstly predefines a set of cluster landmarks that are equal to the number of known cell types for the unannotated dataset. Subsequently, unlabeled cells are assigned to the cluster of the closest target landmark in the embedding space. The assigned cell clusters are matched to annotated cell-type landmarks in the annotated dataset, identifying those with uncertain matching as rare cell types.

**scBalance.**  scBalance (Cheng et al., 2023) is a framework specifically designed for the RCI task. It combines weight sampling and sparse neural networks to emphasize minor cell types without disrupting the annotation efficiency of the major cell populations. scBalance outperforms in handling imbalanced datasets, thus we use it as a benchmark for the RCI task.

**CopyKAT.**  CopyKAT (Gao et al., 2021) serves as a baseline method for the TCD task. It performs hierarchical clustering to categorize cells into clusters according to their estimated gene copy number profiles. Clusters that are significantly enriched in predefined highly confident normal spots in the enrichment analysis (P-value $\leq 0.05$) are designated as normal cells and others as tumors.

**CaSee.**  CaSee (Sh et al., 2022), tailored for distinguishing tumors from normal cells, pre-training the model on a vast of bulk RNA-seq data, followed by employing transfer learning to detect tumors in scRNA-seq data processed by a capsule network. CaSee is a baseline method for the TCD task in this study.

**ikarus.**  ikarus (Dohmen et al., 2022) is designed for detecting tumors from normal cells at the SC level. It first identifies a comprehensive tumor cell signature in the form of a gene set by consolidating abundant annotated datasets. It then employs a logistic regression classifier for stringent discrimination of tumor and normal cells, supplemented by a network-based propagation of cell labels using a custom-built cell-cell network. We leverage ikarus as a baseline for the TCD task in this study.

### C.3  MODEL ARCHITECTURE AND TRAINING DETAILS

In the single-cell model, the number of bins is set to 10. Therefore, the SC expression is processed into 12 different types of tokens: 10 bin tokens, a zero value token, and a special mask token. Then, the dimensions of SC expression embeddings are set the same as text embeddings for addition by the broadcasting mechanism. The main network of the SC Model is a stack of multiple layers of Transformer Encoders. We set it to 12 layers of Transformer Encoders, with 12 attention heads in

each layer. In mask learning, we set $p_m = 0.15$ for masking bin tokens and $p_s = 0.1$ for randomly replacing tokens with others.

For the LLM, ZerOmics uses LLaMA2-13B by default. Each SC embedding is mapped to 500 LLM tokens through a linear layer (tokenizer) and inserted into the instruction. We select three representative and important SC analysis tasks: cell type annotation, rare cell identification, and tumor cell discovery. For each task, we also independently design three instruction templates. During fine-tuning, each cell is randomly assigned a task and a corresponding instruction. In the mixture of LoRAs, we design universal and three task-specific LoRAs, each with the same structure, using $r = 8$ and $\alpha = r$.

For both pre-training and instruction tuning stages, we set the total batch size to 64, which means the pre-GPU batch size is 8. We use AdamW as the optimizer and the learning rate warmup and cosine decay strategies are also used in both stages. The learning rates of the two stages are different, the former is set to 1e-4, and the latter is set to 5e-4.

## C.4 SOURCE CODE

All of the code for this paper, including ZerOmics, its variant models, and most of the pre-training and fine-tuning weights, can be released once the paper is accepted.

## D EXAMPLE ITEMS IN GENE CORPUS

Some gene text summaries are presented in Table 6.

## E INSTRUCTION EXAMPLES

In the instruction tuning stage, ZerOmics employs three instruction tuning tasks. In the evaluation (also inference) stage, ZerOmics employs these three tasks with one additional unseen task. ZerOmics addresses these three shared tasks with similar instruction templates. In summary, the used instruction templates are presented in Table 7.

Table 6: Examples of gene summary items in the gene corpus. Text indicating the associated disease is marked in blue, text describing the associated biological processes is marked in orange, and text indicating associated other genes is marked in green.

| Gene Symbol | GeneCards Summary |
|---|---|
| NXN | NXN (Nucleoredoxin) is a Protein Coding gene. Diseases associated with NXN include Robinow Syndrome, Autosomal Recessive 2 and Autosomal Recessive Robinow Syndrome. Gene Ontology (GO) annotations related to this gene include oxidoreductase activity and thioredoxin-disulfide reductase (NADPH) activity. An important paralog of this gene is NXNL2. |
| TNF | TNF (Tumor Necrosis Factor) is a Protein Coding gene. Diseases associated with TNF include Asthma and Malaria. Among its related pathways are MIF Mediated Glucocorticoid Regulation and TNFR1 Pathway. Gene Ontology (GO) annotations related to this gene include identical protein binding and cytokine activity. An important paralog of this gene is TNFSF15. |
| MEG3 | MEG3 (Maternally Expressed 3) is an RNA Gene, and is affiliated with the lncRNA class. Diseases associated with MEG3 include Kagami-Ogata Syndrome and Liver Disease. |
| SFTA3 | SFTA3 (Surfactant Associated 3) is an RNA Gene, and is affiliated with the lncRNA class. Diseases associated with SFTA3 include Hereditary Ataxia and Choreoathetosis And Congenital Hypothyroidism With Or Without Pulmonary Dysfunction. |
| FRAXE | FRAXE (Fragile Site, Folic Acid Type, Rare, Fra(X)(Q28) E) is a Functional Element gene. Diseases associated with FRAXE include Intellectual Developmental Disorder, X-Linked 109 and Fraxe Intellectual Disability. |
| NXF5 | NXF5 (Nuclear RNA Export Factor 5) is a Pseudogene. Diseases associated with NXF5 include Focal Segmental Glomerulosclerosis 1 and Focal Segmental Glomerulosclerosis. Gene Ontology (GO) annotations related to this gene include RNA binding and nucleotide binding. |

Table 7: Instruction examples for three instruction tuning tasks and one zero-shot task. The content that needs to be inserted in the instruction is marked in cyan.

| Task | Instruction |
|---|---|
| CTA | Given a single-cell token embedding: [SC token], which is derived from the study with the following abstract: [abstract], classify this single-cell token into one of the following categories: [list categories].
Please only respond with one of the categories. |
| RCI | Given a single-cell token embedding: [SC token], which is derived from the study with the following abstract: [abstract], identify whether this single-cell token belongs to a rare cell type.
Please only respond with "Yes" or "No". |
| TCD | Given a single-cell token embedding: [SC token], which is derived from the study with the following abstract: [abstract], determine whether this single-cell token belongs to the following diseases: [list diseases]. Note that it may also be a normal cell.
Please only respond with one of the diseases or "normal cell". |
| CPI | Given a single-cell token embedding: [SC token], which is derived from the study with the following abstract: [abstract], infer which one or more of the following pathways it may be involved in: [list pathways].
Please only respond with one or more of the pathways. |

