# OpenReview forum: "ZerOmics: Toward General Models for Single-Cell Analysis with Instruction Tuning"
_ICLR.cc/2025/Conference — Submitted to ICLR 2025_

### Official Review · Reviewer_gwhJ · 2024-11-02

**Soundness:** 3
**Presentation:** 3
**Contribution:** 4
**Rating:** 6
**Confidence:** 4

**Summary:**

The paper introduces ZerOmics, a novel zero-shot approach that leverages large language models (LLMs) to tackle single-cell analysis tasks without requiring downstream task-specific data. ZerOmics emphasizes a dual-alignment strategy to align single-cell (SC) expression data with gene corpus text embeddings, enabling a broader, zero-shot capability. Results indicate that ZerOmics matches or exceeds performance from established supervised and fine-tuned methods, showcasing notable generalizability.

**Strengths:**

The idea of incorporating gene text embedding is intuitive.

Utilizing LLMs in single-cell data analysis is an important open problem which has attracted a lot of attention recently and ZerOmics provides a promising technical roadmap.

ZerOmics demonstrates strong experimental performance across diverse SC tasks, indicating a high level of effectiveness.

**Weaknesses:**

1. The ablation study on gene text embedding can be more thorough.

2. ZerOmics currently focuses on tasks like cell type annotation, which may not fully highlight the model's zero-shot potential across tasks requiring diverse output formats. It would be helpful to explore the model’s performance on tasks less dependent on natural language formulations, such as regression tasks.

**Questions:**

1. In the pre-training stage, how can the model predict the gene items? How does the model handle instances where multiple masked genes share identical expression bins?

2. Can the author explore the performance of ZerOmics with other types of gene embeddings, such as those obtained from existing single-cell foundation models?

3. Related to the last question, What is the necessity of the first scmodel pre-training part? Could a simpler model, like a single-layer MLP tokenizer, generate single-cell embeddings from expression values directly? More ablation on this could clarify the SC model’s role.

4. How does the author choose the 10 samples for few shot learning? Does the diversity of samples affect the model's performance?

5. Does ZerOmics perform well in scenarios where task objectives extend beyond natural language-compatible tasks, such as regression? Testing on such tasks would demonstrate broader applicability.

6. A recent study (https://www.biorxiv.org/content/10.1101/2023.10.16.562533) addresses similar issues in zero-shot single-cell analysis. Could the authors provide a comparative analysis, covering both methodological differences and performance on shared tasks?

---

> ### Author Response · Authors · 2024-11-26
>
> We appreciate your suggestive comments. Hope the following reply can address your concern.
>
> ***
> **Prediction of the gene items**
>
> In the pre-training stage, the prediction of gene items involves two components: expression bin prediction and gene summary prediction. These tasks are handled through two specialized predictors using fused SC embeddings $Z_{SC}$. Notably, multiple masked genes may share the same expression bin, but they calculate CE loss with different ground truths and are thus distinguishable.
>
> ***
> **Gene embeddings**
>
> Following your suggestion, we tested two representative gene embeddings generated by foundation models, Geneformer [1] and GeneCompass [2], as variant models in the new ablation study (see Section 4.4). Our findings reveal that while these embeddings significantly outperform Gene2vec, they still exhibit a decline in performance compared to the original model. We speculate that this is due to ZerOmics leveraging LLMs as general task processors with advanced reasoning and interpretative capabilities for gene summaries in textual format. This highlights the advantage of utilizing text-based representations in conjunction with LLMs for enhanced task performance.
>
> ***
> **Effectiveness of SC Model & GenePT**
>
> To illustrate the role of the SC Model, we conducted a new ablation study by replacing it with a simpler model, GenePT [3], as a competing variant. GenePT generates gene text embeddings using the LLM and represents each cell by weighting these embeddings with expression values, bypassing the extraction of gene text summaries by the more complex SC Model. Our findings indicate that while the variant model shows a slight performance decline in RCI and TCD tasks, it exhibits a significant decline in the CTA task. This suggests that the SC Model retains an advantage in extracting cell-specific features. Notably, similar designs to the SC Model are employed in two prominent foundation models, scBERT[4] and scGPT [5].
>
> ***
> **Sample selection in the few-shot settings**
>
> In the few-shot learning scenario for the CPI task, we randomly selected 1, 5, and 10 samples from the type 2 datasets (refer to Section 4.1 for details). These type 2 datasets are primarily utilized for training and fine-tuning baseline methods. While they are unavailable to ZerOmics in the zero-shot setting, they are partially accessible in the few-shot settings. The objective of these few-shot settings is to "teach" the LLMs the nature of an unseen CPI task and clarify the expected response. Consequently, sample diversity is less critical under this scenario.
>
>
> ***
> **References**
>
> [1] Theodoris, et al. Transfer learning enables predictions in network biology. 2023.
>
> [2] Yang, et al. GeneCompass: deciphering universal gene regulatory mechanisms with a knowledge-informed cross-species foundation model. 2024.
>
> [3] Chen and Zou. GenePT: a simple but effective foundation model for genes and cells built from chatgpt. 2024.
>
> [4] Yang, et al. scBERT as a large-scale pretrained deep language model for cell type annotation of single-cell rna-seq data. 2022.
>
> [5] Cui, et al. scGPT: toward building a foundation model for single-cell multi-omics using generative AI. 2024.

---

> > ### Comment · Reviewer_gwhJ · 2024-11-27
> > **Thanks for the responses**
> >
> > Thank you for your detailed response, which has addressed most of my concerns. I still have two minor comments for further clarification:
> >
> > 1. You mentioned that your SC model has an advantage in extracting cell features. Could you demonstrate this by presenting a cell UMAP or utilizing other clustering metrics on existing data? This would provide a clearer illustration of the SC model's strengths.
> > 2. Regarding sample selection, given that you have 41 hallmark pathways, I think it may still be important how to choose the few-shot samples. How does the model perform when 10 samples are selected from the same pathway label compared to when the 10 samples are chosen from different labels? Such a comparison would effectively highlight the value of the LLM.

---

### Official Review · Reviewer_1RvP · 2024-11-02

**Soundness:** 2
**Presentation:** 2
**Contribution:** 2
**Rating:** 3
**Confidence:** 4

**Summary:**

The authors present ZerOmics, a zero-shot method to guide LLMs to assist with single-cell (SC) tasks. The authors propose to combine SC expression data with textual information (gene corpus) via summation, to produce embeddings with more information content. They then insert these embeddings into a set of instructions used to finetune an LLM (LLaMA2-13B) on different SC tasks: cell type annotation (CTA), rare cell identification (RCI) and tumor cell identification (TCD). Finetuning is carried out using a mixture of universal and task-specific LoRAs. Results suggest that ZerOmics improves the cell classification tasks across most of the investigated datasets. The authors provide ablation on different parts of their architecture (Gene text embedding, SC tokenizer, Mixture of LoRAs), as well as comparison to different baseline methods for each task.

**Strengths:**

The authors introduce an architecture which combines SC embeddings with textual information about the genes/datasets, in an effort to produce more informative, robust embeddings, which could help alleviate the natural heterogeneity in SC, the well-know batch effect issues and the high-dimensional nature of SC by introducing constraints via additional textual information. I think the main novelty here is focusing on using a general purpose LLM for different SC tasks, by using the instruction tuning paradigm where they format tasks as prompts with SC+Text embeddings included in the instruction. The first introductory section of the paper is clear, situating the work well with regards to the relevant literature in my opinion. The author's also investigate results on three different SC tasks (CTA, RCI and TCD), as well as on 9 different datasets, which is a strong point in their favour.

**Weaknesses:**

In my opinion there main weakness of the paper lies in the experimental design, both in the pipeline itself and in the splitting of the datasets:

- I don't think this approach can truly be called zero-shot: from what I understand, there's a first pretraining of the SC model to combine the SC embedding with the Text embeddings (obtained from a frozen LLM). Then there's the instruction-tuning of the LLM on different tasks (CTA, RCI and TCD), where both universal and task specific LoRAs are used. They then test on held-out datasets, but only for tasks and on classes the LLM was explicitly trained for. This is fundamentally different from true zero-shot learning where a model should be able to handle completely new tasks without task-specific training. This contradicts the paper's claim it's generalisable to unseen tasks.

- This raises the issue of memorisation in the model. Because LLaMa is fine-tuned using instructions that include the learned combined SC+Text embeddings and a task description which contains a curated list of possible answers, this experimental design is explicitly showing the model how to combine embedding-answer pair, which sounds more like memorisation where the model learns how to associate certain embedding patterns with labels from the training set, then inferring true meaningful biological relationships.

- The author's claim that using candidate answers encourages the model to compare and contrast different options, but I find this statement ish misleading, as it implies that merely providing candidate answers automatically leads to improved model reasoning through comparison. In reality, LLMs do not inherently engage in comparative reasoning without additional mechanisms specifically designed to facilitate this process. The cited paper Kim et al. 2024 demonstrates that only with an explicit prompting framework—where candidate answers are iteratively evaluated and refined—can such comparative reasoning effectively enhance prediction accuracy. Therefore, to support their claim, the authors should clarify how ZerOmics specifically leverages candidate answers to encourage genuine comparison or else acknowledge that additional prompt engineering would be necessary to achieve this effect.

- In line with the above points, the instruction set contains the actual abstract of the paper where the dataset is described (in addition to the SC+T embedding and task description), which seems like it could be an important source of data leakage. I would like to see how removing the abstract from the instruction prompt changes the results, to check if this might the case or not.

- It's very unclear to me from the text how the datasets are split into train-test. It seems like the authors are training and testing on the same datasets, with the argument that batch effect is so strong this is the same as testing on a completely different dataset. If this is the case, as I think it is from the description, then I strongly disagree. Yes, batch effect can be very strong, but to make the argument that this is equivalent to testing on a completely different dataset then at a minimum I'd like to see embeddings showing me relative distance between the datasets. This contradicts the paper's claim that the method is generalisable to unseen datasets.

- No confidence intervals or other estimates of uncertainty are included. I encourage the author's to include confidence intervals on their results, either through cross validation or bootstrapping.

**Questions:**

Based on the perceived weaknesses I expanded on above, here's a list of questions to be addressed by the authors, as well as some more general comments included below:

Zero-shot claims:

- Could you clearly justify why you are calling this approach "zero-shot" when you use task-specific LoRAs and instruction tuning for each type of task (CTA, RCI, TCD)?
- Can you demonstrate the model performing well on a completely new type of task it wasn't instruction-tuned for? The experiment on cell pathway inference in 4.3 isn't conclusive as you acknowledge the gene corpus used in pre-training explicitly contains descriptions of these biological processes. This could therefore just be another example of memorisation.
- Furthermore, you don't compare your results to LangCell which has zero-shot capability. I would expect to see LangCell's zero-shot performance as the baseline comparison in this instance.

Memorisation issues:

- How do you ensure your model is learning meaningful biological relationships rather than just memorizing embedding-answer pairs during instruction tuning? What controls or ablation studies have you done to demonstrate the model isn't simply pattern matching against its training data?

Comparative reasoning:

- What specific mechanisms in your architecture enables comparative reasoning between candidate answers? Have you performed experiments showing that providing candidate answers improves performance through comparison rather than just constraining the output space?

Data leakage:

- Have you tested model performance without including dataset abstracts in the instruction prompts? What controls have you implemented to ensure the model isn't leveraging information from the abstracts rather than learning from the SC data?

Dataset splitting:

- Can you provide a clear description of your train-test split methodology?
- Can you demonstrate through embedding analysis that the batch effect differences are comparable to true dataset differences?

Ablation:

- Is the ablation done in a zero-shot setting or after fine-tuning? This significantly affects the interpretation of the ablation study results, so needs to be clearly highlighted in the text.

Generalisation:

- Have you tested the model on truly independent datasets from different studies/labs?

Further comments:

- Lines 120 - 121: “Gene expression data are often affected by many non-biological factors and the information they reveal is not as profound as text data”.

I suggest you might want to rephrase this statement, which implies text data contains more information than the actual SC data. Gene expression data is a reflection of the huge biological complexity inherent to our cellular machinery. Furthermore, cell-specific gene expression profiles are more idiosyncratic to a specific cell, whereas textual descriptions tend to be a lot more general, so what exactly is meant by “gene expression data is not as profound as text data” here?

- Line 158 - 161: "Genes that are highly expressed in most cells,
such as housekeeping genes, may exhibit lower expression levels in this context. In contrast, genes that are lowly expressed but crucial for identifying cell states, such as transcription factors, may exhibit higher expression levels."

I believe this statement is factually incorrect: Quantile normalisation doesn't inherently lower the expression value of highly expressed genes or increase the value of lowly expressed ones. Instead, it preserves the ranking between genes in each sample, as well as forcing expression values to follow the same distribution across samples. Could you expand or explain this point?

Line 202 - 206: "Multi-modal mask learning for semantic alignment. Cell expression and gene text embeddings reveal distinct levels of biomedical information within SC data. Using multi-modal learning to align their semantic spaces inspires the model to extract more comprehensive representations. For computational efficiency, ZerOmics utilizes the broadcasting to directly add Z\_T to Z\_E, which is then encoded to the contextual SC embeddings that contain the gene functional semantics, ..."

Am I correct to interpret broadcasting as elementwise summation of the SC and Text embedding? This should be clarified in the text. Can you really call this semantic alignment if what you're doing to align them is to broadcast them together via direct summation?

- Lines 301 - 303: “Three challenging single-cell tasks -- cell type annotation, rare cell identification and tumor cell discovery”

CTA are a staple task that has repeatedly been shown to be accurately predicted even by simple logistic regression based methods (see [1] and [2]). While interesting and important, cell type annotation is not a particularly challenging task so I would rephrase this statement.

- Line 072: "inadequately designed task-specific heads" - why are they inadequately designed? What do you do to design a better task-head? How do you test this better design?

- Line 133: "elevatable" What does this mean?

- As a general comment, I would say the text is quite clear until section 3.2, but the rest is quite confusing to read and I suggest the author's might try to rework these sections to gain clarity.

[1] https://www.biorxiv.org/content/10.1101/2023.10.19.563100v1

[2] https://www.biorxiv.org/content/10.1101/2023.10.16.561085v1

---

> ### Author Response · Authors · 2024-11-27
>
> We thank the reviewer for their valuable feedback.
>
> ***
> **Zero-shot claims**
>
> From the perspective of **model architecture and paradigm**, ZerOmics undoubtedly belongs to the zero-shot setting. The three task-specific LoRAs are utilized in the instruction tuning stage, which is different from fine-tuning output layers with task-specific datasets (see Figure 1). In the fine-tuning paradigm, though most foundation models perform large-scale pre-training, they are still necessary to use task-specific heads on specific datasets for fine-tuning to better adapt to the target task. However, for ZerOmics in a zero-shot setting, both LLM and LoRAs are frozen once the pre-training is complete. It is particularly emphasized that even when applied to unseen tasks, we will not modify the model weights or add new task-specific LoRA. Task-specific LoRAs in the ZerOmics are \textbf{not used for} specific downstream tasks. In addition, instruction tuning is not limited to these three tasks. In this study, we selected these three tasks as representatives from various SC analysis tasks. It is also possible to use other tasks in the instruction tuning stage.
>
> From the perspective of **datasets**, both SC datasets and gene text summaries highlight the zero-shot settings. For the train-test split strategy, please see the following ``Data splitting'' section. For gene text summaries, they convey the information of genes, rather than focusing on cells. Gene-level text does not directly present cell-level information, especially does not highlight the heterogeneity between different cells, thus textual information is more likely to be task-specific. Their applicability to unrelated tasks remains limited. Therefore, there is no information leakage problem, and it still meets the zero-shot setting.
>
> Additionally, we compared LangCell's performance in both zero-shot and few-shot settings on the CTA task. The RCI and TCD tasks were excluded, as there is no prior research utilizing LangCell for these tasks, nor are there generally recognized task-specific heads available for them. For the CPI task, we only evaluated fine-tuned LangCell because we followed the original settings outlined in its paper, which exclusively employed the fine-tuning approach.
>
> ***
> **Generalization**
>
> The evaluation datasets are collected across diverse studies (see Appendix B.2 for details), which are not included in training and instruction tuning databases.
>
> ***
> **Data splitting**
>
> We used **independent** training and evaluation SC databases in our experiments.
>
> In brief, we categorize the SC datasets in our study based on their usages: (1) large-scale SC datasets used for ZerOmics' pre-training and multi-modal instruction tuning stages (see Appendix A.1 for details). (2) Training and fine-tuning datasets split from evaluation datasets used for baseline methods' training, fine-tuning, and few-shot settings of ZerOmics. The detailed train-test split strategy is similar to Langcell, which can be seen in Appendix B.1. This part of the evaluation data is **not available for ZerOmics in zero-shot setting**. (3) Held-out evaluation datasets, which are collected from diverse studies and labs used for testing model performance (see Appendix B.2 for details).
>
> ***
> **Comparative reasoning**
>
> Selecting an optimal result from a large candidate pool is a traditional mechanism called retrieval-based strategy [1]. It has been well-studied by existing work such as [1], [2], [3], [4], [5], [6]. Compared to constraining output layers, the given candidate answer set provides supervised information to guide LLM's learning. Furthermore, users can control the granularity of results by changing the candidate answer sets. For instance, users can add "T cell" to the candidate sets to help LLM handle the coarse-grained annotation task, or add "CD4+ T cell" to the set for solving the cell subtype annotation task.
>
> ***
> **Data leakage**
>
> In our opinion, the dataset description solely provides coarse-grained information about included cell groups, which cannot be utilized independently for cell-level tasks. It helps to shrink the search scope into a subset of the LLM's knowledge pool, thus improving LLM's reasoning efficiency.
>
> ***
> **Writing statements**
>
> For "Further comments" 1, 4, and 6, we have rephrased the statement correspondingly and the main text, please check the newest version of the pdf for details.

---

> ### Author Response · Authors · 2024-11-27
>
> **References**
>
> [1]: Song, et al. Uni-Encoder: A Fast and Accurate Response Selection Paradigm for Generation-Based Dialogue Systems. 2023.
>
> [2]: Tang, et al. GraphGPT: Graph instruction tuning for large language models. 2024.
>
> [3]: Wang, et al. LLMs as zero-shot graph learners: Alignment of GNN representations with LLM token embeddings. 2024.
>
> [4]: Guo, et al. Regiongpt: Towards region understanding vision language model. 2024.
>
> [5]: Dai, et al. InstructBLIP: Towards General-purpose Vision-Language Models with Instruction Tuning. 2023.
>
> [6]: Liu, et al. Visual instruction tuning. 2024.

---

> ### Author Response · Authors · 2024-11-28
> **[Gentle Reminder]: Kind Request for Reviewer's Feedback**
>
> Dear Reviewer 1RvP:
>
> Thank you once again for your valuable and constructive review, which has helped us refine some of the unclear descriptions and further design new ablation studies to address your concerns. The corresponding results and rephrased statements are uploaded to the newest version of the main text, and we have given detailed explanations in the previous comments.
>
> We would like to kindly remind you that the discussion deadline is approaching. After this deadline, we may not have the opportunity to respond to your comments.
>
> We sincerely appreciate your dedication and look forward to your feedback.
>
> Sincerely, ICLR 2025 Conference Submission 2577 Authors

---

> > ### Comment · Reviewer_1RvP · 2024-12-01
> >
> > I thank the authors for their response, I have gone over their revised manuscript and read their responses to reviewers. I still have concerns about the experimental design, which I feel have not been adequately addressed. They are as follows:
> >
> > **Data splitting**
> >
> > The author response about data independence remains problematic. While they claim to use "independent training and evaluation SC databases", their paper states that type 2 (training/fine-tuning) and type 3 (evaluation) datasets "are split from the same evaluation datasets, sharing the same sampling conditions and sequencing processes." The authors argue this is valid because these splits "approximately satisfy the same distribution assumption" but this directly contradicts the claim of independence, highlighting my original concern. To demonstrate true independence and generalization capability, evaluation should be performed on datasets from completely separate experiments with different sampling conditions. Without clear metrics quantifying dataset independence or clear evidence that evaluation datasets truly come from different experimental conditions, the authors haven't adequately addressed my core concern about whether their method genuinely generalizes across datasets. This ambiguity in their data splitting methodology makes it impossible to verify their claims about zero-shot generalization performance.
> >
> > **Zero-shot claims**
> >
> > I remain unconvinced by the authors explanation of their zero-shot claims. While they argue this is "undoubtedly" zero-shot because they freeze the LoRAs after instruction tuning, this sidesteps the fundamental issue that they are still training task-specific components (LoRAs) for CTA, RCI, and TCD tasks, and then evaluating on datasets that share "sampling conditions and sequencing processes" with their training data. So, yes, in a zero-shot setting task specific output layers are fine-tuned, but this seems fundamentally different from training task-specific LoRAs during instruction tuning and then testing model performance on these tasks and on data with similar distributions. A true zero-shot approach should be evaluating model performance on new tasks without any task-specific training of the model, to evaluate if the model has learnt inherent, transferable 'biological' information.
> >
> > **Data leakage**
> >
> > The authors' state that dataset abstracts are merely "coarse-grained information" that "cannot be utilized independently for cell-level tasks", which significantly understates their information content. Scientific abstracts frequently contain explicit descriptions of cell type markers, expression patterns associated with specific cell populations, and relationships between gene expression and cellular phenotypes. For instance, an abstract studying T cell populations might state "CD4+ T cells were identified by high expression of CD4 and IL7R." This directly links cell type identity to specific gene expression patterns. Similarly, abstracts often describe how cell types were validated using specific marker genes, or detail expression-based criteria used for cell type assignment. Given that this information is provided in the instruction prompt alongside the cell's expression data, the model could potentially learn to match these text-described patterns rather than discovering true biological relationships from the expression data alone. This makes it impossible to determine whether the model is actually learning to understand single-cell data or is simply exploiting detailed biological knowledge provided in the abstracts. The authors' claim that abstracts merely "help shrink the search scope" understates this serious methodological concern.
> >
> > **Memorization**
> >
> > The ablation results in Table 5 actually strengthen my concerns about memorization and the model's reliance on text descriptions rather than learning true biological patterns. The model performs significantly better when using both abstracts and task-specific components (85.56\% accuracy on CTA-PBMC68K) compared to using just gene embeddings (79.71\% with GeneCompass, 65.85\% with Gene2vec). Moreover, removing either the text components or task-specific LoRAs leads to substantial performance drops. This pattern suggests the model's strong performance depends heavily on matching text descriptions with expression patterns during the instruction tuning phase, rather than learning to understand the underlying biology.
> >
> > Overall, while I believe this is a valuable avenue of investigation, the author response has not changed my opinion with regards to the methodological concerns I have on the experimental design. Therefore I am maintaining my rating.

---

### Official Review · Reviewer_cF5L · 2024-11-06

**Soundness:** 2
**Presentation:** 2
**Contribution:** 2
**Rating:** 5
**Confidence:** 5

**Summary:**

This work presents a method to encode single cell expression data as a token to an LLM enabling it to be finetuned with these new tokens for tasks such as cell type classification. The single cell sample representation is learned by a denoising autoencoder and then this representation is linearly transformed into the dimension of a token embedding for a language model.

**Strengths:**

This work tackles an interesting problem of enabling language models to reason about the modality of single cell genomics. The experiments performed compare against supervised baselines.
There are many ablation studies performed to explore what aspects of the method are working.

**Weaknesses:**

I expected to see more exploration into the method of incorporating a single cell observation. This work only discusses a single approach (besides the gene2vec ablation) while I'm sure the research team experienced many configurations that didn't work. It would be nice to have experiments with these alternatives as well.

Edit: After reviewing comments from Reviewer 1RvP I agree that the method does not appear to be zero-shot due to pre-training for each task as confirmed by the statement "To assess the performance in a zero-shot setting, SC datasets are split into a training set for pre-training and tuning, and a test set for evaluation".

I now believe a re-framing of the paper should occur to prevent confusion. The method is not as simple as taking the SC sample embedding for a single cell and putting it in a prompt as the current framing implies.

**Questions:**

None

---

> ### Comment · Reviewer_cF5L · 2024-11-22
> **Downgrade**
>
> After reviewing comments from Reviewer 1RvP I agree that the method does not appear to be zero-shot due to pre-training for each task as confirmed by the statement "To assess the performance in a zero-shot setting, SC datasets are split into a training set for pre-training and tuning, and a test set for evaluation".
>
> I now believe a re-framing of the paper should occur to prevent confusion. The method is not as simple as taking the SC sample embedding for a single cell and putting it in a prompt as the current framing implies.
>
> In light of this I will change my score.

---

> > ### Author Response · Authors · 2024-11-27
> >
> > Thanks for your suggestive comments and valuable opinions.
> >
> > ***
> > **Gene embeddings**
> >
> > Following your suggestion, we tested two representative gene embeddings generated by foundation models, Geneformer and GeneCompass, as variant models in the new ablation study (see Section 4.4). Our findings reveal that while these embeddings significantly outperform Gene2vec, they still exhibit a decline in performance compared to the original model. We speculate that this is due to ZerOmics leveraging LLMs as general task processors with advanced reasoning and interpretative capabilities for gene summaries in textual format. This highlights the advantage of utilizing text-based representations in conjunction with LLMs for enhanced task performance.
> >
> > ***
> > **Zero-shot**
> >
> > We apologize for the inaccuracies in the description. To further clarify the dataset used in this article, we modified the original description as follows:
> >
> > We categorize the SC datasets in our study based on their usages: (1) large-scale SC datasets used for ZerOmics' pre-training and multi-modal instruction tuning stages (see Appendix A.1 for details). (2) Training and fine-tuning datasets split from evaluation datasets used for baseline methods' training, fine-tuning, and few-shot settings of ZerOmics. The detailed train-test split strategy is similar to Langcell, which can be seen in Appendix B.1. This part of the evaluation data is **not available for ZerOmics in zero-shot setting**. (3) Held-out evaluation datasets, which are collected from diverse studies and labs used for testing model performance (see Appendix B.2 for details).
> >
> > Additionally, from the perspective of model architecture and paradigm, ZerOmics undoubtedly belongs to the zero-shot setting. The three task-specific LoRAs are utilized in the instruction tuning stage, which is different from fine-tuning output layers with task-specific datasets (see Figure 1). In the fine-tuning paradigm, even if large-scale pre-training is used in foundation models, it is still necessary to use task-specific heads on specific datasets for fine-tuning to better adapt to the target task. However, for ZerOmics in a zero-shot setting, once large-scale pre-training is complete, both LLM and LoRA are frozen. It is particularly emphasized that even when applied to unseen tasks, we will not modify the model weights or add new task-specific LoRA. Task-specific LoRAs in the ZerOmics are \textbf{not used for} specific downstream tasks. In addition, instruction tuning is not limited to these three tasks. In this study, we selected these three tasks as representatives from various SC analysis tasks. It is also possible to use other tasks in the instruction tuning stage.
> >
> > ***
> > We sincerely hope that you will re-read our response and the revised manuscript. In fact, ZerOmics is exactly what you thought it was, "taking the SC sample embedding for a single cell and putting it in a prompt as the current framing implies."

---

> ### Author Response · Authors · 2024-11-28
> **[Gentle Reminder]: Kind Request for Reviewer's Feedback**
>
> Dear Reviewer cF5L:
>
> Thanks for your positive comments and valuable opinions. To address your recent proposed concerns, we have explained the zero-shot claims in the previous comments and designed new ablation studies to further demonstrate ZerOmics' superiority in extracting sing-cell embeddings. The corresponding results and rephrased statements are uploaded to the newest version of the main text.
>
> We would like to kindly remind you that the discussion deadline is approaching. After this deadline, we may not have the opportunity to respond to your comments.
>
> We sincerely appreciate your dedication and look forward to your feedback.
>
> Sincerely, ICLR 2025 Conference Submission 2577 Authors

---

### Author Response · Authors · 2024-11-22
**Responses to Common Questions and Concerns**

We sincerely appreciate the suggestive comments from all reviewers. In response to the issues that multiple reviewers are concerned about, we first provide a unified reply as follows:

***
**Zero-shot claims**

From the perspective of **model architecture and paradigm**, ZerOmics undoubtedly belongs to the zero-shot setting. The three task-specific LoRAs are utilized in the instruction tuning stage, which is different from fine-tuning output layers with task-specific datasets (see Figure 1). In the fine-tuning paradigm, even if large-scale pre-training is used in foundation models, it is still necessary to use task-specific heads on specific datasets for fine-tuning to better adapt to the target task. However, for ZerOmics in a zero-shot setting, once large-scale pre-training is complete, both LLM and LoRA are frozen. It is particularly emphasized that even when applied to unseen tasks, we will not modify the model weights or add new task-specific LoRA. Task-specific LoRAs in the ZerOmics are **not used for** specific downstream tasks. In addition, instruction tuning is not limited to these three tasks. In this study, we selected these three tasks as representatives from various SC analysis tasks. It is also possible to use other tasks in the instruction tuning stage.

From the perspective of **datasets**, both SC datasets and gene text summaries highlight the zero-shot settings. For the train-test split strategy, please see the following “Data splitting” section. For gene text summaries, they just convey the information of genes, rather than focusing on cells. Gene-level text does not directly present cell-level information, especially does not highlight the heterogeneity between different cells, thus textual information is more likely to be task-specific. Their applicability to unrelated tasks remains limited. Therefore, there is no information leakage problem, and it still meets the zero-shot setting.

***
**Data splitting**

We used **independent** training and evaluation SC databases in our experiments.

In brief, we categorize the SC datasets in our study based on their usages: (1) large-scale SC datasets used for ZerOmics' pre-training and multi-modal instruction tuning stages (see Appendix A.1 for details). (2) Training and fine-tuning datasets split from evaluation datasets used for baseline methods' training, fine-tuning, and few-shot settings of ZerOmics. The detailed train-test split strategy is similar to Langcell, which can be seen in Appendix B.1. This part of the evaluation data is **not available for ZerOmics in zero-shot setting**. (3) Held-out evaluation datasets, which are collected from diverse studies and labs used for testing model performance (see Appendix B.2 for details).

***
**Ablation studies**

After carefully reading the reviewer's suggestions, we realized that the original ablation studies were not clear enough and did not fully address the issues that the reviewer was concerned about. Therefore, we designed a new ablation study based on the reviewers' concerns and immediately carried out a series of experiments. In the new ablation studies, we give ablations on gene features, SC models, and LoRAs. First, we use gene embeddings generated by gene2vec, gene-former, and GeneCompass as gene features to replace the original text summaries to explore the impact of different gene embeddings (especially those generated by foundation models). We then replace the SC model with GenePT, which also uses gene text summaries and a variant model that removes the linear tokenizer to further clarify the role of the SC model. The ablation of LoRAs remains the same, but the analysis content of the text is modified. Finally, whether it is the original ablation studies or the modified ones, it is worth emphasizing that we trained these variant models from scratch instead of simply replacing or removing them from instruction-tuned ZerOmics.

***
Revised manuscript (including new ablation studies) has been updated in the PDF.

---

### Meta-Review · Area_Chair_FR4L · 2024-12-22

**Metareview:**

The ZerOmics paper presents a zero-shot single-cell analysis using large language models (LLMs) and gene text embeddings, showing strong generalizability and outperformance of established methods in tasks like cell type annotation, rare cell identification, and tumor cell discovery. However, several reviewers raised concerns about the true "zero-shot" nature of the method, highlighting its dependence on pre-training and task-specific fine-tuning. In the initial review, additional ablation studies, clarification of the pre-training role, and testing on a wider range of tasks, such as regression, were recommended. Although the authors provided additional results and revised the manuscript to address these concerns, most reviewers remained hesitant to raise their scores during the rebuttal phase.  Overall, the current version does not seem to meet the standards of ICLR.

**Additional Comments On Reviewer Discussion:**

Additional concerns regarding data leakage, unclear dataset splitting, and the need for comparative analysis with similar methods were also raised. However, these issues were not adequately addressed after the rebuttal, and the authors did not get to provide further clarification about some of the above issues before the deadline. Overall, the current version fails to meet the standards required for ICLR.

---

### Decision · Program_Chairs · 2025-01-22

Reject